# Efficient Revocable Attribute-Based Encryption with Data Integrity and Key Escrow-Free

**Meijuan Huang** [1], **Yutian Liu** [1], **Bo Yang** [2,*], **Yanqi Zhao** [3] and **Mingrui Zhang** [4]

1 School of Mathematics and Information Science, Baoji University of Arts and Sciences, Baoji 721013, China; huangmeijuan@bjwlxy.edu.cn (M.H.); liuyutian@stu.bjwlxy.edu.cn (Y.L.)
2 School of Computer Science, Shaanxi Normal University, Xi'an 710119, China
3 School of Cyberspace Security, Xi'an University of Posts and Telecommunications, Xi'an 710121, China; zhaoyanqi2021@xupt.edu.cn
4 Software Engineering Institute, East China Normal University, Shanghai 200062, China; zhangmingrui@stu.ecnu.edu.cn
* Correspondence: byang@snnu.edu.cn

**Abstract:** Revocable attribute-based encryption (RABE) provides greater flexibility and fine-grained access control for data sharing. However, the revocation process for most RABE schemes today is performed by the cloud storage provider (CSP). Since the CSP is an honest and curious third party, there is no guarantee that the plaintext data corresponding to the new ciphertext after revocation is the same as the original plaintext data. In addition, most attribute-based encryption schemes suffer from issues related to key escrow. To overcome the aforementioned issues, we present an efficient RABE scheme that supports data integrity while also addressing the key escrow issue. We demonstrate the security for our system, which is reduced to the decisional q-parallel bilinear Diffie-Hellman exponent (q-PBDHE) assumption and discrete logarithm (DL) assumption. The performance analysis illustrates that our scheme is efficient.

**Keywords:** attribute-based encryption; cloud computing; data integrity; key escrow

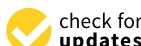



## 1. Introduction

Cloud storage services provide major advantages in data management as data continues to grow and digitization processes accelerate, and more and more companies and individuals are choosing to employ cloud storage services to satisfy their data storage demands. Compared with traditional local storage, cloud storage has the advantages of high storage efficiency, high scalability, and low management overhead. However, cloud storage providers (CSP) may attempt to access sensitive data, which can lead to potential privacy risks [1–3]. The key to solving this problem is to store the data in ciphertext. The traditional method can only achieve one-to-one sharing. If the file is shared with several users, it must be encrypted multiple times, which lacks flexibility and fine-grained access control. Attribute-based encryption (ABE) [4] technology effectively solves this problem; it can provide file confidentiality and a one-to-many sharing mechanism over encrypted data. Data in an ABE scheme is encrypted using access policy. The user can decrypt and achieve plaintext when the user's attributes match the access policy in the ciphertext. Therefore, the user fully utilizes cloud storage services to maintain data security and privacy. Ciphertext-policy attribute-based encryption (CP-ABE) [5] and key-policy attribute-based encryption (KP-ABE) [6] are two types of ABE. In CP-ABE, the user's attribute set corresponds to the key, and the access policy corresponds to the ciphertext, while the opposite is true for KP-ABE. The user can decrypt only when the attributes match the access policy.

Currently, CP-ABE is widely used in healthcare, financial services, e-commerce, and other scenarios, but in many practical application scenarios, CP-ABE is still confronted with numerous challenges, such as user revocation [7] and key escrow issues. Revocable

attribute-based encryption restricts access to data by controlling user attributes such as job titles or security clearance levels. It allows data owners to revoke access to certain users when necessary, thus providing greater flexibility and fine-grained control over data sharing, enabling greater data security and privacy.

### 1.1. Related Works

More and more programs are now focusing on the issue of revocation. Pirretti et al. [8] developed a revocable encryption scheme that supports indirect revocation, where each attribute in the scheme contains a valid time range and the authority periodically updates the attribute and redistributes the user's key information. Li et al. [9] constructed a revocable scheme that introduces the concept of user groups to achieve efficient user revocation, where the group administrator updates the keys of unrevoked users when any user leaves, and the scheme outsources part of the computation to the CSP to reduce the user computation burden. In [10], an efficient direct RABE scheme was provided. In the scheme, a user revocation list and a time interval are added. The revoked users are added to the revocation list and can not decrypt the ciphertext after the key time expires, and the key of the unrevoked users will be updated. Xiang et al. [11] adopted version control technology to support real-time revocation and the private key for the unrevoked user is updated by the subset covering technique. In [12], the data owner does not need to be online during the revocation process, but the unrevoked user is required to update the decryption key frequently, and the data storage center needs to re-encrypt the ciphertext, which is computationally intensive and not suitable for resource-constrained environments. Xiong et al. [13] combined revocable encryption with cloud-assisted IoT, where the trusted authority center manages a user revocation list. The identities and current time nodes of these users will be added to the list once they have been deleted from the system. Using key update parameters generated by the trusted authority center, users who are not revoked will update their own decryption keys. Lan et al. [14] constructed an efficient revocable ABE scheme with rich attribute representation. The proxy server is in charge of partial decryption and receives a conversion key from the key generating center. When a user's attributes change or he or she is deleted from the system, both the decryption key and the conversion key for unrevoked users need to be updated. The above scheme achieves revocation by maintaining a revocation list or updating the key periodically, but the length of the list increases with the rapid change of personnel flow, and this method requires the user to update the key frequently online at any time, which has a large computational overhead.

Sahai et al. [15] introduced the ciphertext delegation technique, in which the cloud server achieves user revocation by re-encrypting the ciphertext, but the scheme cannot be applied in CP-ABE. In [16], a server-assisted RABE scheme is constructed, in which the ciphertext should be converted by the CSP using the relevant conversion key, and if the user is removed from the system, the CSP will no longer be able to help him or her to convert the ciphertext. The CP-ABE scheme in [17] applied a modular ciphertext delegation method that allows third parties to convert ciphertexts under a stricter policy, enabling user revocation. Ma et al. [18] constructed a revocable, secure data deletion and authentication CP-ABE scheme. The scheme uses attribute association trees to reconstruct new access policy and re-encrypts ciphertext data when a user is deleted, so that the deleted user is unable to decrypt the new ciphertext. In [19], a traceable RABE scheme is constructed by uploading the revocation list along with the ciphertext to CSP. When a user revokes from the system, the CSP updates the ciphertext using the update key transmitted by the authorization center, and the user's identity is related to the leaf node to achieve user tracking. In [20], the CSP re-encrypts the ciphertext by combining the original ciphertext with the updated material broadcast by the authority center using the ciphertext delegation algorithm. These schemes use the CSP to update the ciphertext to achieve revocation, which saves computing resources to a certain extent. However, since the proxy third-party server is honest and curious, there is no guarantee that the plaintext data corresponding to

the new ciphertext after revocation is consistent with the original plaintext data, which is what we call the data integrity issue. Aiming to resolve this problem, Ge et al. [21] used a user-verifiable approach to construct a new RABE scheme that supports data integrity. Based on Waters' scheme [22], they encrypted both the plaintext data and a random value, allowing the user to check the consistency for the plaintext data. However, the scheme has a key escrow problem.

In addition, ABE schemes also come with the key escrow problem. In the traditional ABE scheme, the key generation center (KGC) generates the decryption keys for the users, which means that the KGC has the ability to access and decrypt data. To overcome this issue, the schemes in [23,24] generate decryption keys for users by introducing multiple authorization centers, each of which can only calculate partial keys. The scheme in [25] used an unmanaged key issue protocol executed between the CSP and the KGC, but the computational cost is too high. The scheme in [26] used an unmanaged key issue protocol executed between the KGC and the user, solving the key escrow issue effectively. Recently, some novel ABE schemes were presented, such as CP-ABE with shared decryption [27], ABE with privacy protection and accountability [28], multi-authority CP-ABE [29,30], and revocable blockchain-aided ABE [31].

Therefore, to address the integrity issue and the key escrow issue in revocation, we constructed an efficient revocable ABE scheme that supports data integrity and solves the key escrow issue. The specific contributions are as follows:

- **Data integrity:** Under the new access policy, when the CSP performs the revocation operation to generate the ciphertext, the user can check whether the plaintext corresponding to the new ciphertext is the same as the original encrypted plaintext.
- **Key-escrow free**: Attribute authority was introduced, and a secure 2PC protocol is executed between the key authority and the attribute authority to generate the user's private key. Neither side can get the complete private key, which solved the key escrow problem.
- **Security and efficiency**: Based on the assumption of decisional q-PBDHE, our scheme is secure under chosen plaintext attacks. Performance analysis illustrates the practicability and effectiveness of the proposed scheme.

*1.2. Organization*

We review some knowledge about topics like bilinear maps and linear secret sharing in Section 2. We provide an overview of the security model and the system model in Section 3. We present an efficient RABE scheme based on the Waters' scheme in Section 4. Sections 5 and 6 discuss the safety and feasibility of our scheme, respectively. Finally, we summarize our work in Section 7.

## 2. Preliminaries

We focus on describing the specific construction of our RABE scheme, and the notation used in the paper is explained in Table 1.

***Bilinear maps*** The bilinear map $e : G \times G \to G_T$ has the following properties:

- Bilinear: $\forall a, b \in G, u, v \in Z_p^*, e(a^u, b^v) = e(a, b)^{uv}$ holds.
- Non-degeneracy: $e(a, b) \neq 1$.
- Computability: $e(a, b)$ can be effectively calculated.

***Access policy*** The set $A \subseteq 2^{\{P_1, P_2, \cdots, P_n\}}$ is called monotonous if $B \in A$ and $B \subseteq C$, we have $C \in A$. The access policy is the monotone set $A$ in all non-empty subsets for $P$, i.e., $A \subseteq 2^{\{P_1, P_2, \cdots, P_n\}} \setminus \{\varnothing\}$. The sets are referred to as the authorization sets, otherwise, the sets are referred to as the unauthorized sets.

***Linear secret sharing scheme (LSSS)*** A linear secret sharing scheme $\Pi$ on $Z_p$ meets the following two conditions:

- Each participant's share is the component of the vector on $Z_p$.

- Define a share generating matrix $M_{m \times n}$ and for all $j \in [1, m]$, we define a function $\rho(j) : \{1, \cdots, m\} \to \{P_1, P_2, \cdots, P_n\}$, where $1, 2, \cdots, m$ is the number of rows in $M_{m \times n}$. Randomly choosing vector $\vec{u} = (r, u_2, \cdots, u_n)$, where $r \in Z_p$ is a secret shared value, $u_2, \cdots u_n \in Z_p$ was picked randomly. $M \cdot \vec{u}$ represents $m$ secret share values shared according to $\Pi$.

**Table 1.** Symbols Definition.

| Symbol | Description |
|---|---|
| $G, G_T$ | Two multiplicative cyclic groups with prime order $p$ |
| $g$ | A generator in $G$ |
| U | Collection of all system attributes |
| \|U\| | The number of elements of the set U |
| $\mathbb{S}$ | Collection of user attributes |
| $\mathbb{S} \subseteq U$ | $\mathbb{S}$ is a subset of U |
| PPT | Probabilistic polynomial time |
| 2PC | Two-party computing |
| *Param* | Public parameters |
| *MSK* | Master key |
| *SK* | User private key |
| *CT* | Ciphertext |
| $P = \{P_1, P_2, \cdots, P_n\}$ | Participant set |
| $M_{m \times n}$ | A matrix with $m$ rows and $n$ columns |
| $M_j$ | The $j$-th row of $M$ |
| $(M_{m \times n}, \rho)$ | Access policy |
| $[1, m]$ | A set of $1, 2, \cdots, m$ |

LSSS satisfies the linear reconfiguration property that members in the authorization set $\mathbb{S}$ can recover secret as follows: For an access policy $A$, let $\mathbb{S} \in A$ be any authorized set, and let $Q = \{j : \rho(j) \in \mathbb{S}\} \subset \{1, \cdots, m\}$, we can compute the constant set $\{\eta_j\}_{j \in Q}$ in polynomial time using the knowledge of linearity algebra such that $\sum_{j \in Q} \eta_j \zeta_j = r$, where $\zeta_j = \left( M_j \cdot \vec{u} \right)$. In this paper, $(M_{m \times n}, \rho)$ stands for access policy, and s can be recovered only when the attributes of the user meet $(M_{m \times n}, \rho)$.

***Discrete logarithm assumption (DL)*** Let $G$ be a group of prime order $p$, and $g$ be a generator. The DL assumption says, that given $(g, g^\varphi)$ for randomly chosen $\varphi \in Z_p^*$, for the PPT algorithm $\mathcal{A}$, $\Pr[A(g, g^\varphi) = \varphi] \leq \varepsilon$ is negligible.

***Decisional q-Parallel Bilinear Diffie-Hellman Exponent assumption (q-PBDHE)*** Let $a, d_1, \cdots, d_q, r \in Z_p$ be chosen randomly, and $e : G \times G \to G_T$ be a bilinear map. Given tuple:

$$\vec{y} = \left\{ g, g^r, g^a, \cdots, g^{a^q}, g^{a^{q+2}}, \cdots, g^{a^{2q}}, \right.$$
$$\forall_{1 \leq i \leq q} g^{r \cdot d_i}, g^{a/d_i}, \cdots, g^{a^q/d_i}, g^{a^{q+2}/d_i}, \cdots, g^{a^{2q}/d_i}$$
$$\left. \forall_{1 \leq i,l \leq q, l \neq i} g^{a \cdot r \cdot d_l/d_i}, \cdots, g^{a^q \cdot r \cdot d_l/d_i} \right\}$$

The decisional q-PBDHE assumption means that there is no PPT algorithm to distinguish the distribution of $\mathbb{F}_{q-PBDHE} = \left\{ \left( \vec{y}, e(g,g)^{a^{q+1}r} \right) \right\}$ and $\mathbb{R}_{q-PBDHE} = \left\{ \left( \vec{y}, \widetilde{R} \right) \right\}$, where $\widetilde{R}$ be a random element in $G_T$. The decisional q-PBDHE assumption was first defined and proved to be safe in [22].

## 3. System Model

We will give the roles of each entity, the formal definition, and the security model for the RABE scheme.

Our RABE system includes five entities: Data Owner (DO), Data User (DU), Cloud Service Provider (CSP), Key Authority (KA), and Attribute Authority (AA), which is illustrated in Figure 1.

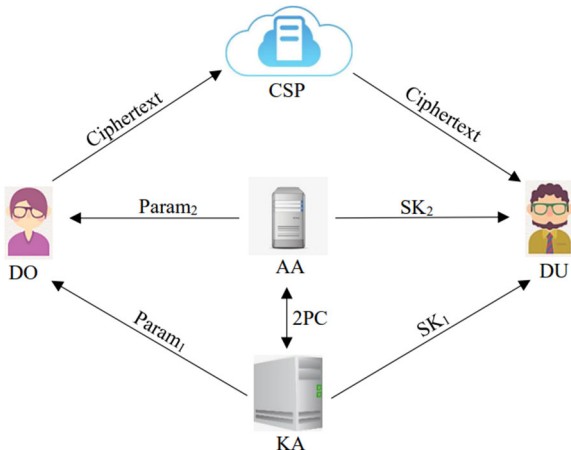

**Figure 1.** System structure for RABE.

**DO**: The DO sets an access policy for the data, generates file ciphertext using a combination of symmetric encryption (AES) and the CP-ABE algorithm, and finally sends the complete ciphertext to the CSP.

**CSP**: The CSP stores ciphertext uploaded by the DO and performs the revocation operation.

**DU**: The DU downloads ciphertext from the CSP. If the attributes of the DU match the access policy embedded in the ciphertext, he or she can decrypt the data to obtain plaintext.

**KA/AA**: The KA and AA are responsible for system initialization and generating user private keys.

### 3.1. Formal Definition

The algorithms in the RABE scheme are as below:

(1) $Setup\_KA(\lambda, U) \rightarrow (Param_1, MSK_1)$. This algorithm generates the public key $Param_1$ and private key $MSK_1$ of the KA according to the security parameter $\lambda$ and system attribute set U.

(2) $Setup\_AA(Param_1) \rightarrow (Param_2, MSK_2)$. This algorithm generates the public key $Param_2$ and private key $MSK_2$ of the AA according to $Param_1$.

(3) $Keygen(MSK_1, MSK_2, Param, \mathbb{S}) \rightarrow SK$. This algorithm generates the user's private key $SK$ through a secure 2PC protocol.

(4) $Encrypt(Param, F, (M_{m \times n}, \rho)) \rightarrow CT$. This algorithm encrypts data files $F$ and uploads the ciphertext to the CSP.

(5) $Decrypt_{or}(SK, CT) \rightarrow F$. This algorithm inputs $SK$ and $CT$, and outputs a shared data file $F$ or a special symbol $\perp$.

(6) $Revoke(CT, (\overline{M}_{\overline{m} \times \overline{n}}, \overline{\rho})) \rightarrow CT'$. This algorithm inputs $CT$ and a revocation access policy $(\overline{M}_{\overline{m} \times \overline{n}}, \overline{\rho})$, and it outputs a revoked ciphertext $CT'$.

(7) $Decrypt_{re}(SK', CT, CT') \rightarrow F$. This algorithm inputs updated private key $SK'$, $CT$ and $CT'$, and outputs a shared data file $F$ or a special symbol $\perp$.

### 3.2. Security Model

We define two security models for the RABE scheme, namely the selective plaintext attack and the data integrity attack. These are described through the interactive attack games (**Game-I** and **Game-II**) between adversary $\mathcal{A}$ and challenger $\mathcal{C}$.

**Game-I** describes a security game under selective plaintext attack.

- Initialization: $\mathcal{A}$ chooses a challenge access policy $(M^*_{m^* \times n^*}, \rho^*)$ and sends it to challenger $\mathcal{C}$.

- Setup: $\mathcal{C}$ executes the *Setup* algorithm to obtain the master public key *Param* and returns it to $\mathcal{A}$.
- Private key query phase 1: $\mathcal{A}$ chooses a user attribute set $\mathbb{S}$, which requires that $\mathbb{S}$ cannot meet $(M^*{}_{m^* \times n^*}, \rho^*)$. $\mathcal{C}$ runs the *Keygen*, and generates the private key *SK* and returns it to $\mathcal{A}$.
- Challenge: $\mathcal{A}$ chooses two data files $F_0$ and $F_1$ of equal length to $\mathcal{C}$. $\mathcal{C}$ chooses $\theta \in \{0,1\}$ randomly and encrypts $F_\theta$ to get the challenge ciphertext $CT^*$. $\mathcal{C}$ returns the ciphertext $CT^*$ to $\mathcal{A}$.
- Private key query phase 2: Similar to the previous stage, $\mathcal{C}$ continues to answer $\mathcal{A}$'s query.
- Guess: $\mathcal{A}$ outputs its guess $\theta' \in \{0,1\}$ for $\theta$.

  We define $\mathcal{A}$'s advantage in the above game as $Adv = \left| \Pr[\theta' = \theta] - \frac{1}{2} \right|$.

**Definition 1.** *Our RABE scheme is selective plaintext attack secure, if for all PPT adversary $\mathcal{A}$, the advantage $Adv = \left| Pr[\theta' = \theta] - \frac{1}{2} \right|$ is negligible.*

**Game-II** describes a security game under data integrity attack.

- Setup: $\mathcal{C}$ executes *Setup* algorithm to get public parameter *Param* and returns it to $\mathcal{A}$.
- Private key query phase 1: $\mathcal{A}$ can perform the key extraction query on the user attribute set $\mathbb{S}$. $\mathcal{C}$ returns *SK* to $\mathcal{A}$ by executing the *Keygen* algorithm.
- Challenge: $\mathcal{A}$ sends the data file $F$ and a challenge access policy $(M_{m \times n}, \rho)$ to $\mathcal{C}$. Then $\mathcal{C}$ sends challenge ciphertext $CT$ to $\mathcal{A}$ by executing the *Encrypt* algorithm.
- Private key query phase 2: Similar with the previous stage, $\mathcal{C}$ continues to answer $\mathcal{A}$'s query.
- Guess: $\mathcal{A}$ outputs attribute set $\mathbb{S}'$ and revoked ciphertext $CT'$. $\mathcal{A}$ wins the integrity game if $Dec_{re}(SK_{\mathbb{S}'}, CT, CT') \notin \{F, \perp\}$.

  We define $\Pr[\mathcal{A}wins]$ to represent the adversary $\mathcal{A}'$s advantage in the above game.

**Definition 2.** *The proposed scheme achieves the data integrity of ciphertext after revocation if for all PPT adversary $\mathcal{A}$, the advantage $Pr[\mathcal{A}wins]$ is negligible.*

## 4. Our RABE Construction

(1) $Setup\_KA(\lambda, U) \to (Param_1, MSK_1)$. This algorithm inputs system security parameter $\lambda$, and attribute set U, generates two cyclic groups $G$, $G_T$ with prime order $p$ and bilinear map $e : G \times G \to G_T$. Let $g$ be a generator in $G$. The KA randomly selects $g, \mu, \nu \in G$, $a, b, \alpha_1 \in Z_p^*$, hash function $\hat{H} : G_T \to Z_p^*$ and $h_1, h_2, \cdots, h_{|U|}$, then the algorithm outputs

$$Param_1 = \left( G, G_T, e, g, g^a, \mu, \nu, \{h_i | i = 1, 2, \cdots, |U|\}, g^b, E^{\alpha_1}, \hat{H} \right), MSK_1 = (\alpha_1, b).$$

The KA publishes $Param_1$ and keeps $MSK_1$ secretly, where $E = e(g, g)$.

(2) $Setup\_AA(Param_1) \to (Param_2, MSK_2)$. The AA selects $\alpha_2 \in Z_p^*$ randomly, outputs $Param_2 = (E^{\alpha_2})$, $MSK_2 = (\alpha_2)$. The AA keeps $MSK_2$ secretly and publishes $Param_2$. Then we have

$$Param = \left( G, G_T, e, g, g^a, \mu, \nu, \{h_i | i = 1, 2, \cdots, |U|\}, g^b, E^\alpha, \hat{H} \right), MSK = (\alpha_1, \alpha_2, b),$$

where $\alpha = \alpha_1 + \alpha_2$.

(3) $KeyGen(MSK_1, MSK_2, Param, \mathbb{S}) \to SK$. In this algorithm, the KA and the AA use the secure 2PC protocol to generate the user's private key. Firstly, the KA inputs $(\alpha_1, b)$, the AA inputs $\alpha_2$, the protocol computes $\omega = (\alpha_1 + \alpha_2)b$ and returns $\omega$ to the AA, where the KA does not know $\alpha_2$ and the AA does not know $(\alpha_1, b)$, then the AA and the KA interact to generate $SK_2$:

- The AA selects $t_1 \in Z_p^*$ at random, the AA computes $X_1 = g^{\omega/t_1} = g^{(\alpha_1 + \alpha_2)b/t_1}$, and generates the knowledge proof of $\omega, t_1$, then sends $X_1$ and $PoK(\omega, t_1)$ to the KA.

- The KA selects $s, \tau \in Z_p^*$ at random, computes $T_1 = X_1^{\tau/b} = g^{(\alpha_1 + \alpha_2)\tau/t_1}$, $T_2 = g^{s\tau \cdot a}$, then transmits $T_1, T_2$ and $PoK(\tau, s, b)$ to the AA.

- The AA selects $t_2 \in Z_p^*$ at random, computes $X_2 = (T_1^{t_1} T_2)^{t_2} = (g^{(\alpha_1 + \alpha_2)\tau} g^{s\tau a})^{t_2}$, then sends $X_2$ and $PoK(t_2)$ to KA.

- The KA computes $T_3 = X_2^{1/\tau} = (g^{(\alpha_1 + \alpha_2)} g^{sa})^{t_2}$, sends $PoK(\tau)$ and $T_3$ to the AA.

- The AA calculates $D = T_3^{1/t_2} = g^{\alpha} g^{sa}$, and then the AA transmits $SK_2 = \{D = g^{\alpha} g^{sa}\}$ to the DU.

- The KA computes $D_0 = g^s, D_x = h_x{}^s, \forall x \in \mathbb{S}$ and sends $SK_1 = \{D_0 = g^s, D_x = h_x{}^s\}$ to the DU.

- The DU's final private key is $SK = \{D = g^{\alpha} g^{sa}, D_0 = g^s, D_x = h_x{}^s (\forall x \in \mathbb{S})\}$. The above protocol is illustrated in Figure 2.

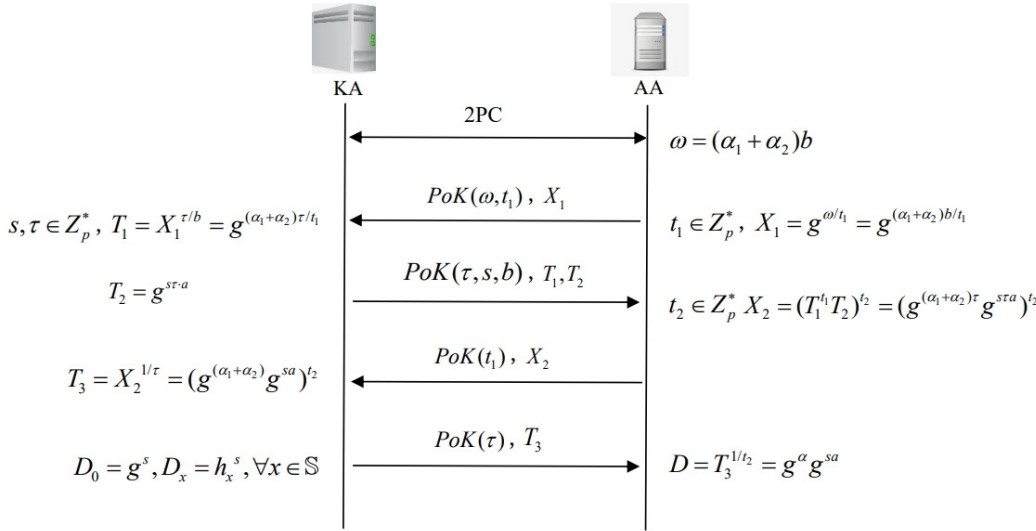

**Figure 2.** The proposed key issuing protocol.

(4) $Encrypt(Param, F, (M_{m \times n}, \rho)) \to CT$. This algorithm inputs the shared data file $F$, $Param = \left( G, G_T, e, g, g^a, \mu, \nu, \{h_i | i = 1, 2, \cdots, |U|\}, g^b, E^{\alpha}, \hat{H} \right)$ and access policy $(M_{m \times n}, \rho)$, for each row of $M_{m \times n}$, the function $\rho$ associates rows of $M_{m \times n}$ to attributes, which is $\rho : \{1, 2, \cdots, m\} \to U$. The algorithm encrypts the file $F$ using the AES algorithm, then gets the shared data ciphertext $CF = Enc_{ck}(F)$, where $ck$ is a symmetric key. The DO selects a vector $\vec{u} = (r, u_2, \cdots, u_n) \in Z_p^*$, $c_j \in Z_p$ randomly, computes $\zeta_j = \vec{u} \cdot M_j, j \in [1, m]$. Then

$$C_1 = ck \cdot E^{\alpha r}, C_2 = g^r, C_{3,j} = h_{\rho(j)}^{-c_j} g^{a\zeta_j}, C_{4,j} = g^{c_j}, \forall j \in [1, m], C_5 = \mu^{\hat{H}(F)} \nu^{\hat{H}(ck)},$$

Let $C = ((M_{m \times n}, \rho), C_1, C_2, C_{3,j}, C_{4,j}, C_5, j \in [1, m])$, then the DO sends $CT = \{CF, C\}$ to the CSP for storage.

(5) $Decrypt_{or}(SK, CT) \to F$. The DU runs the algorithm and decrypts the ciphertext $CT$. The algorithm inputs private key $SK = \{D, D_0, D_x(\forall x \in \mathbb{S})\}$, $CT = \{CF, C\}$. If the attribute set $\mathbb{S}$ satisfies $(M_{m \times n}, \rho)$, lets $Q = \{j : \rho(j) \in \mathbb{S}\} \subset \{1, \cdots, m\}$, calculates the constant $\{\eta_j\}_{j \in Q}$ such that $\Sigma_{j \in Q} \eta_j M_j = (1, 0, 0, \cdots, 0)$, the algorithm computes

$$ck = C_1 \Big/ \frac{e(D, C_2)}{(\Pi_{j \in Q} e(D_0, C_{3,j}) \cdot e(D_{\rho(j)}, C_{4,j}))^{\eta_j}}.$$

Then checks if $C_5 = \mu^{\hat{H}(F)} v^{\hat{H}(ck)}$, outputs $ck$ and decrypts the shared file $F$ further. Otherwise, outputs $\perp$. If $\mathbb{S}$ does not satisfy $(M_{m \times n}, \rho)$, decryption fails.

(6) $Revoke(CT, (\overline{M}_{\overline{m} \times \overline{n}}, \overline{\rho})) \to CT'$. The CSP runs the algorithm. It inputs $CT = \{CF, C\}$, a revocation access policy $(\overline{M}_{\overline{m} \times \overline{n}}, \overline{\rho})$, and for each row of $\overline{M}_{\overline{m} \times \overline{n}}$, defines the function $\overline{\rho} : \{1, 2, \cdots, \overline{m}\} \to U$. It outputs a revoked ciphertext $CT'$ under a revoked access policy $(M'_{m' \times n'}, \rho')$, where $\rho' : \{1, 2, \cdots, m'\} \to U$, $m' = m + \overline{m}$, $n' = n + \overline{n}$. Then, it randomly selects $\overrightarrow{\widetilde{u}} = (\widetilde{r}, \widetilde{u}_2, \cdots, \widetilde{u}_{n'}) \in Z_p^{n'}$ and $\widetilde{c}_j \in Z_p$ for each $j \in [1, m']$, computes $\widetilde{\zeta}_j = \overrightarrow{\widetilde{u}} \cdot M'_j, j \in [1, m']$. The algorithm computes $\hat{C}$:

$$L_1 = C_1, L_2 = C_2, L_{3,j} = C_{3,j}, L_{4,j} = C_{4,j}, j \in [1, m], L_{3,j} = 1_G, L_{4,j} = 1_G, j \in [m+1, m'],$$

where $1_G$ is the identity element of $G$. Then the algorithm computes $\widetilde{C}$:

$$K_1 = E^{\alpha \widetilde{r}}, K_2 = g^{\widetilde{r}}, K_{3,j} = g^{a\widetilde{\zeta}_j} h_{\rho(j)}^{-\widetilde{c}_j}, K_{4,j} = g^{\widetilde{c}_j}, \forall j \in [1, m'].$$

And computes $C'$:

$$C'_1 = L_1 \cdot K_1, C'_2 = L_2 \cdot K_2, C'_{3,j} = L_{3,j} \cdot K_{3,j}, C'_{4,j} = L_{4,j} \cdot K_{4,j}, \forall j \in [1, m'], C'_5 = C_5.$$

Let $C' = \left( (M'_{m' \times n'}, \rho'), C'_1, C'_2, C'_{3,j}, C'_{4,j}, C'_5, j \in [1, m'] \right)$, outputs $CT' = \{C', CF\}$.

(7) $Decrypt_{re}(SK', CT, CT') \to F$. The algorithm inputs $SK'$, $CT = \{CF, C\}$ and $CT' = \{C', CF\}$, verifies whether $C'_5 = C_5$, if not, outputs $\perp$. Then, if the set of attribute $\mathbb{S}'$ of $SK'$ meets $(M', \rho')$, let $Q' = \{j : \rho'(j) \in \mathbb{S}'\} \subset \{1, \cdots, m'\}$, and there is a constant $\left\{ \eta'_j \right\}_{j \in Q'}$ such that $\Sigma_{j \in Q'} \eta'_j \cdot M'_j = (1, 0, 0, \cdots, 0)$. Then the DU computes:

$$ck = C'_1 / \frac{e(D, C'_2)}{(\Pi_{j \in Q'} e(D_0, C'_{3,j}) \cdot e(D_{\rho'(j)}, C'_{4,j}))^{\eta'_j}},$$

otherwise, outputs $\perp$. Finally, checks if $C'_5 = \mu^{\hat{H}(F)} v^{\hat{H}(ck)}$, outputs $ck$, and decrypts the shared file $F$ further. Otherwise, outputs $\perp$.

Kim et al. [17] proved that $(M', \rho')$ is a valid access policy with respect to a LSSS scheme. Therefore $CT'$ is a valid revoked ciphertext.

## 5. Scheme Analysis

### 5.1. Correctness Analysis

In $Decrypt_{or}$ algorithm:

$$\begin{aligned}
&\frac{e(D, C_2)}{(\Pi_{j \in Q} e(D_0, C_{3,j}) \cdot e(D_{\rho(j)}, C_{4,j}))^{\eta_j}} \\
&= \frac{e(g^\alpha g^{sa}, g^r)}{(\Pi_{j \in Q} e(g^s, g^{a\zeta_j} h_{\rho(j)}^{-c_j}) \cdot e(h_{\rho(j)}^s, g^{c_j}))^{\eta_j}} \\
&= \frac{E^{\alpha r} \cdot E^{sar}}{E^{sa \cdot \Sigma_{j \in Q} \zeta_j \eta_j}} \\
&= E^{\alpha r}
\end{aligned}$$

### 5.2. Security Analysis

**Theorem 1.** *Assuming that the decisional q-PBDHE assumption holds, then our RABE construction described above is semantic secure under chosen plaintext attack.*

**Proof.** Assume a PPT adversary $\mathcal{A}$ exists with a non-negligible advantage to break the security for our RABE construction, so we construct a polynomial time simulator $\mathcal{S}$ using $\mathcal{A}$ to break the decisional q-PBDHE assumption.

- Init. $\mathcal{S}$ picks a bilinear map $e : G \times G \to G_T$, and $a, d_1, \cdots, d_q, r \in Z_p$ randomly. $\mathcal{S}$ exposes:

$$
\begin{aligned}
\vec{y} = \Big\{ &g, g^r, g^a, \cdots, g^{a^q}, g^{a^{q+2}}, \cdots, g^{a^{2q}}, \\
\forall_{1 \le i \le q} &g^{r \cdot d_i}, g^{a/d_i}, \cdots, g^{a^q/d_i}, g^{a^{q+2}/d_i}, \cdots, g^{a^{2q}/d_i} \\
&\forall_{1 \le i, l \le q, l \ne i} g^{a \cdot r \cdot d_l/d_i}, \cdots, g^{a^q \cdot r \cdot d_l/d_i} \Big\}.
\end{aligned}
$$

$\mathcal{S}$ randomly selects $\sigma \in \{0, 1\}$, if $\sigma = 0$, take $Z = E^{a^{q+1}r}$, let $T = \left( \vec{y}, Z \right)$; if $\sigma = 1$, take $Z \in G_T$ and let $T = \left( \vec{y}, Z \right)$, $\mathcal{A}$ picks a challenge access policy $(M^*{}_{m^* \times n^*}, \rho^*)$ to $\mathcal{S}$.

- Setup. $\mathcal{S}$ picks $\alpha' \in Z_p$ randomly, computes $E^\alpha = E^{a \cdot a^q} \cdot E^{\alpha'}$. This implicitly sets $\alpha = \alpha' + a^{q+1}$. $\mathcal{S}$ orchestrates group element $h_1, h_2, \cdots, h_{|U|}$ as follows: For attributes $1 \le x \le |U|$, $\mathcal{S}$ chooses a value $w_x$ at random, let $Y$ be the set of $j$ such that $\rho^*(j) = x$. $\mathcal{S}$ sets $h_x$ as

$$
h_x = g^{w_x} \prod_{j \in Y} \prod_{1 \le k \le n^*} g^{\frac{a^k M^*_{j,k}}{d_j}}.
$$

Because of the randomness of $g^{w_x}$, $h_x$ is distributed randomly. If $Y = \varnothing$, then $h_x = g^{w_x}$. The simulator $\mathcal{S}$ chooses a hash function $\hat{H}$ and $\mu, v \in G$ randomly, returns the public parameters

$$
Param = \left\{ G, G_T, e, g, g^a, \mu, v, \{h_x | 1 \le x \le |U|\}, g^b, E^\alpha, \hat{H} \right\}
$$

to $\mathcal{A}$.

- Private key query phase 1. $\mathcal{A}$ submits attribute set $\mathbb{S}$, where $\mathbb{S}$ does not satisfy $M^*{}_{m^* \times n^*}$. Simulator $\mathcal{S}$ chooses $t \in Z_p$ at random and finds the vector $\vec{\eta} = (\eta_1, \eta_2, \cdots, \eta_{n^*}) \in Z_p^{n^*}$ such that $\eta_1 = -1$. For $\{j : \rho^*(j) \in \mathbb{S}\}$, we have $\vec{\eta} \cdot M_j^* = 0$. $\mathcal{S}$ computes

$$
D_0 = g^t \prod_{1 \le j \le n^*} \left( g^{a^{q+1-j}} \right)^{\eta_j} = g^s,
$$

thus, implicitly defining

$$
s = t + \eta_1 a^q + \eta_2 a^{q-1} + \eta_{n^*} a^{q-(n^*-1)}.
$$

By defining $s$ so that $g^{as}$ contains $g^{-a^{q+1}}$, the unknown term $g^\alpha$ can be eliminated when constructing $D$. $\mathcal{S}$ computes

$$
D = g^{\alpha'} g^{at} \prod_{j=2}^{n^*} \left( g^{a^{q+2-j}} \right)^{\eta_j}.
$$

Now compute $D_x$ for $x \in \mathbb{S}$. If there is no $j$ that makes $\rho^*(j) = x$, then $D_x = D_0{}^{w_x}$; if there is multiple $j$ that makes $\rho^*(j) = x$, since $\mathcal{S}$ cannot simulate $g^{a^{q+1}/d_j}$, it is necessary to ensure that the expression for $D_x$ does not contain terms shaped like $g^{a^{q+1}/d_j}$.

Because $\vec{\eta} \cdot M_j^* = 0$, everything in this form can be cancelled. Let $Y = \{j, \rho^*(j) = x\}$ and calculate

$$
D_x = D_0{}^{w_x} \prod_{j \in Y} \prod_{i=1}^{n*} \left( g^{(a^i/d_j)t} \prod_{\substack{l = 1, \cdots, n* \\ l \neq i}} \left(g^{a^{q+1+i-l}/d_j}\right)^{\eta_l} \right)^{M_{j,i}^*} .
$$

The simulator $\mathcal{S}$ returns $SK = \{D, D_0, D_x(\forall x \in \mathbb{S})\}$ to $\mathcal{A}$.

- Challenge. $\mathcal{A}$ selects two messages $F_0$ and $F_1$ of equal length. Simulator $\mathcal{S}$ chooses a coin $\theta \in \{0,1\}$ randomly and encrypts the file $F_\theta$ using the AES algorithm to generate the shared data ciphertext $CF = Enc_{ck}(F_\theta)$, where $ck$ is a symmetric key, then $C_1 = ck \cdot Z \cdot e(g^r, g^{\alpha'}), C_2 = g^r$. $\mathcal{S}$ chooses

$$
\vec{u} = (r, ra + u_2', ra^2 + u_3', \cdots, ra^{n-1} + u_{n*}') \in Z_p^{n*},
$$

where $u_2', \cdots, u_{n*}' \in Z_p$ randomly, $r$ is the secret value to be shared. In addition, $\mathcal{S}$ chooses $t_1', t_2', \cdots, t_m' \in Z_p$, $C_5 \in G$ at random, we define $R_j$ to be the set of all $l$ satisfying $l \neq j$ such that $\rho^*(j) = \rho^*(l)$, $j = 1, 2, \cdots, n^*$, compute

$$
\begin{cases}
C_{3,j} = h_{\rho^*(j)}^{t_j'} \left(g^{d_j \cdot r}\right)^{-w_{\rho^*(j)}} \left( \prod_{2 \leq i \leq n*} (g^a)^{M_{j,i}^* y_i'} \right) \cdot \left( \prod_{l \in R_j} \prod_{1 \leq i \leq n*} \left(g^{a^i \cdot r \cdot (d_j/d_l)}\right)^{M_{l,i}^*} \right) , \\
C_{4,j} = g^{-rd_j} g^{t_j'}
\end{cases}
$$

$C = (C_1, C_2, C_{3,j}, C_{4,j}, C_5, j \in [1, m])$. The simulator returns $CT = \{CF, C\}$ to $\mathcal{A}$.

- Private key query phase 2. Similar with the previous stage, $\mathcal{S}$ continues to answer $\mathcal{A}$'s query.
- Guess. $\mathcal{A}$ outputs guess $\theta' \in \{0, 1\}$ of $\theta$. $\mathcal{S}$ outputs $\sigma' = 0$ when $\theta' = \theta$, it means $T \in \mathbb{F}_{q-PBDHE}$; $\mathcal{S}$ outputs $\sigma' = 1$ when $\theta' \neq \theta$, it means $T \in \mathbb{R}_{q-PBDHE}$. When $\sigma = 1$, $\mathcal{A}$ does not obtain any information from $\theta$, so $\Pr[\theta' \neq \theta | \sigma = 1] = \frac{1}{2}$. When $\theta' \neq \theta$, $\mathcal{S}$ guesses $\sigma' = 1$, $\Pr[\sigma' = \sigma | \sigma = 1] = \frac{1}{2}$. When $\sigma = 0$, $\mathcal{A}$ knows the ciphertext of $F_\theta$, because the advantage of $\mathcal{A}$ is $\varepsilon$, $\Pr[\theta' = \theta | \sigma = 0] = \frac{1}{2} + \varepsilon$. When $\theta' = \theta$, $\mathcal{S}$ guesses $\sigma' = 0$, $\Pr[\sigma' = \sigma | \sigma = 0] = \frac{1}{2} + \varepsilon$. The advantages of $\mathcal{S}$ obtained from the above are

$$
\frac{1}{2} \Pr[\sigma' = \sigma | \sigma = 0] - \frac{1}{2} \Pr[\sigma' = \sigma | \sigma = 1] = \frac{1}{2} \left( \frac{1}{2} + \varepsilon \right) - \frac{1}{2} \times \frac{1}{2} = \frac{\varepsilon}{2}.
$$

Therefore, Theorem 1 holds. □

**Theorem 2.** *The proposed scheme supports data integrity under the DL assumption.*

**Proof.** Assume a PPT adversary $\mathcal{A}$ exists with a non-negligible advantage to break the security for our RABE construction, so we can construct a polynomial time simulator $\mathcal{S}$ using $\mathcal{A}$ to break the DL assumption.

- Setup. $\mathcal{S}$ obtains a discrete logarithmic tuple $(G, G_T, p, g, g^\varphi)$, and $\mathcal{S}$ attempts to compute the value $\varphi$. $\mathcal{S}$ generates public parameters through the following steps. $\mathcal{S}$ sets a bilinear map $e : G \times G \to G_T$, selects $h_1, \cdots, h_{|U|} \in G$, $\alpha, a, b, \gamma \in Z_p$, and computes $g^a, g^b, E^\alpha, \mu = g^\varphi, \nu = g^\gamma$. $\mathcal{S}$ picks hash function $\hat{H} : G_T \to Z_p$ at random, and returns

$$
Param = \left( G, G_T, e, g, g^a, \mu, \nu, \{h_i | i = 1, 2, \cdots, |U|\}, g^b, E^\alpha, \hat{H} \right)
$$

to adversary $\mathcal{A}$.

- Private key query phase 1. $\mathcal{S}$ selects an attribute set $\mathbb{S}$, and executes *KeyGen(MSK, Param,$\mathbb{S}$)* $\rightarrow SK$ and returns $SK$ to $\mathcal{A}$.
- Challenge. $\mathcal{A}$ submits $F$ and a challenge access policy $(M, \rho)$ to $\mathcal{S}$. $\mathcal{S}$ execute *Encrypt(Param,F,($M_{m \times n}, \rho$))* $\rightarrow CT = \{CF, C\}$, where $C_5 = \mu^{\hat{H}(F)}\nu^{\hat{H}(ck)}$, $CF = Enc(F, ck)$, $C = ((M_{m \times n}, \rho), C_1, C_2, C_{3,j}, C_{4,j}, C_5, j \in [1, m])$. $\mathcal{S}$ returns $CT$ to $\mathcal{A}$.
- Private key query phase 2. Similar to the previous stage, $\mathcal{S}$ continues to answer $\mathcal{A}$'s query.
- Output. $\mathcal{A}$ outputs a revoked ciphertext $CT' = \{CF', C'\}$, where $CF' = Enc(F', ck')$, $C' = \left((M'_{m' \times n'}, \rho'), C'_1, C'_2, C'_{3,j}, C'_{4,j}, C'_5, j \in [1, m']\right)$. $\mathcal{A}$ wins if $F' \notin \{F, \bot\}$ and $C_5 = C'_5$.

If $\mathcal{A}$ wins, the simulator $\mathcal{S}$ selects the attribute set $\mathbb{S}'$ that meets access policy $(M'_{m' \times n'}, \rho')$. $\mathcal{S}$ generates the private key $SK_{\mathbb{S}'}$, decrypts the ciphertext $CT'$ to get the symmetric key $ck'$, and then gets the $F'$. According to $C_5 = C'_5 \Leftrightarrow \nu^{\hat{H}(ck)}\mu^{\hat{H}(F)} = \nu^{\hat{H}(ck')}\mu^{\hat{H}(F')}$, $\mathcal{S}$ computes $\varphi \cdot (\hat{H}(F) - \hat{H}(F')) = \gamma \cdot (\hat{H}(ck') - \hat{H}(ck))$. Since $F' \notin \{F, \bot\}$, so that means $\hat{H}(F) \neq \hat{H}(F')$. Finally, the simulator $\mathcal{S}$ gets $\varphi$.

Therefore, Theorem 2 holds. $\square$

## 6. Performance Analysis

The performance of our scheme is analyzed in terms of functionality, computational cost, and experimental perspectives.

### 6.1. Functional Analysis

The functional analysis between our scheme and the schemes in [21,24,26] is shown in Table 2. None of the comparison schemes can simultaneously meet the three functional requirements listed in the table, that is, cannot simultaneously meet integrity, key escrow, and revocation. The scheme in this paper can simultaneously meet the above three functional requirements and adopts LSSS with strong expression ability as the access policy. Therefore, from the perspective of functionality, our scheme is more suitable for practical application.

**Table 2.** Functionality.

| Scheme | Integrity | Key-Escrow Free | User Revocation | Access Policy |
|---|---|---|---|---|
| [21] | √ | × | √ | LSSS |
| [24] | × | √ | × | LSSS |
| [26] | × | √ | √ | Tree |
| Ours | √ | √ | √ | LSSS |

### 6.2. Computation Analysis

In this section, we compare our scheme with other schemes in terms of calculated costs, as shown in Table 3. It can be seen from Table 3, in the key generation phase, the computational cost required by our scheme is consistent with that in [21,24], and lower than that of [26]. In the encryption stage, our scheme has more advantages than those in [21,24,26]. At the decryption stage, the computational cost of our scheme is consistent with reference [24] and lower than [21,26]. In the revocation phase, the calculation cost of our scheme is lower than that of [21], which is almost the same as that of [26]. In general, the approach in this paper has a low computing overhead, where $m$ represents the number of rows of the matrix in LSSS, $y$ represents the number of leaf nodes in the access tree, both $m$ and $y$ correspond to the number of attributes, so their meanings in Table 3 are the same.

In Table 3, $u$ represents the number of attributes for the user, $m$ represents the number of rows of the matrix in LSSS, $y$ represents the number of leaf nodes in the access tree, $E_1$ represents exponential operations in group $G$, $E_T$ represents exponential operations in group $G_T$, $P$ represents bilinear pair operation.

**Table 3.** Calculations cost.

| Scheme | Key Generation | Encryption | Decryption | Revocation |
|---|---|---|---|---|
| [21] | $(u+3)E_1$ | $(6m+4)E_1 + 2E_T + 2P$ | $10E_T + 10P$ | $(12m+6)E_1 + 4E_T + 4P$ |
| [24] | $(u+3)E_1$ | $(4m+1)E_1 + E_T + P$ | $5E_T + 5P$ | – |
| [26] | $(2u+8)E_1$ | $(2y+4)E_1 + 2E_T + 2P$ | $8E_T + 8P$ | $(2y+4)E_1 + 2E_T + 2P$ |
| Ours | $(u+3)E_1$ | $(3m+3)E_1 + E_T + P$ | $5E_T + 5P$ | $(6m+4)E_1 + 2E_T + 2P$ |

*6.3. Experimental Analysis*

In this section, in order to better evaluate the performance of our scheme, we conducted simulation experiments between our scheme and the scheme in reference [21] (abbreviated as RI-CP-ABE). The experimental environment configuration is as follows: AMD Ryzen 5 5600U with Radeon Graphics 2.30 GHz, 16.0 GB RAM, Windows 10 operating system. Our scheme used the IntelliJIDEA2018 tool, jPBC2.0 open-source encryption library, and we selected a Type A elliptic curve with group order bit length of 512 bits for the experiment, the expression is $y^2 = x^3 + x$. We used JAVA language for programming, and the LSSS access matrix is programmed in the form of a binary tree.

We conducted simulation experiments in the aspects of system establishment time, key generation time, encryption time, decryption time, revocation time, and decryption after revocation time. Since only scheme RI-CP-ABE has integrity, therefore, we compared our scheme with RI-CP-ABE. The specific algorithm of reference in RI-CP-ABE is shown in Appendix A. Because our computer runs with limited memory, the number of attributes in the access policy is set to 4, 8, 16, and 32 (the number of attributes in the system). The experiment was conducted 100 times in total, and the average value of the experimental results of 100 times was taken as the final result of this experiment to ensure the accuracy of the experiment.

The time cost of system setup is shown in Figure 3, indicating that the calculation cost of the method in this article is basically the same as that described in the literature RI-CP-ABE. The system key generation time overhead is illustrated in Figure 4. The results of experimental simulations demonstrate that the calculation cost in our scheme is more than that of the literature in RI-CP-ABE. Because we introduced the 2PC protocol to solve the key escrow issue, which guards against the misuse of users' private keys, it is more useful in real-world applications. The figure shows that the time growth rates of the two systems are nearly equal as the number of attributes increases. Furthermore, the key is generated only once, and the impact on the overall system efficiency can be ignored.

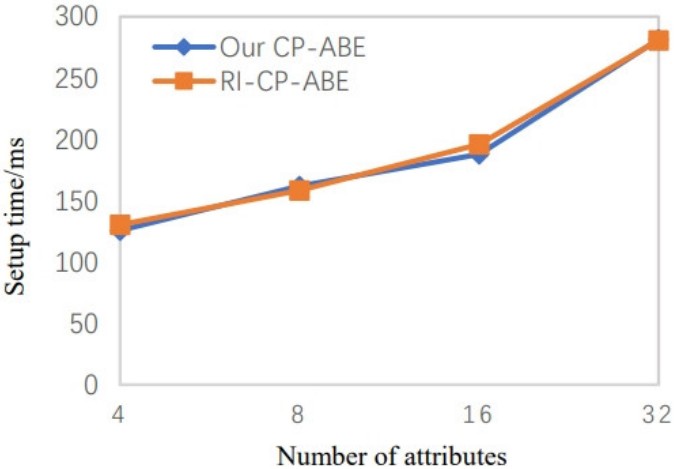

**Figure 3.** Setup time when the number of attributes increases [21].

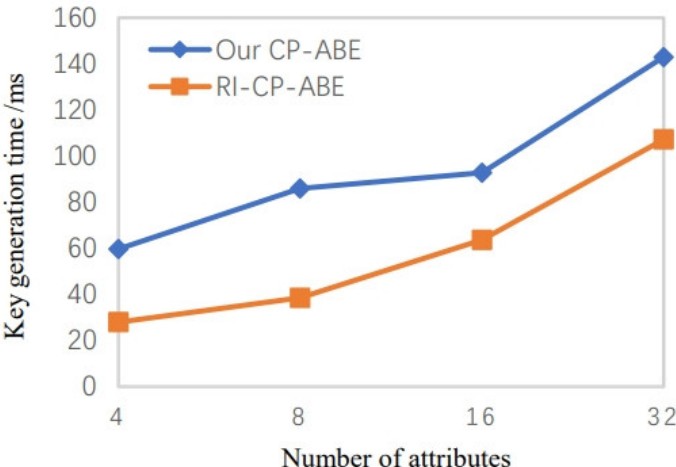

**Figure 4.** Key generation time when the number of attributes increases [21].

The system encryption time and the initial decryption time overhead are shown in Figures 5 and 6. The figures demonstrate that compared to the technique in RI-CP-ABE, ours takes much less encryption and decryption time. Therefore, our scheme significantly reduces the computing burden on users.

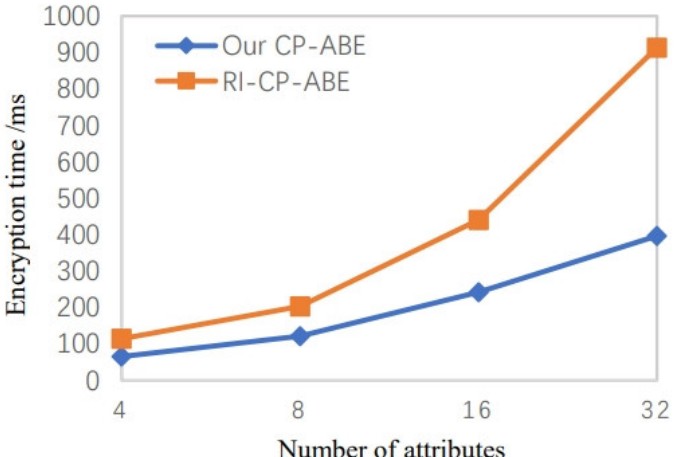

**Figure 5.** Encryption time when the number of attributes increases [21].

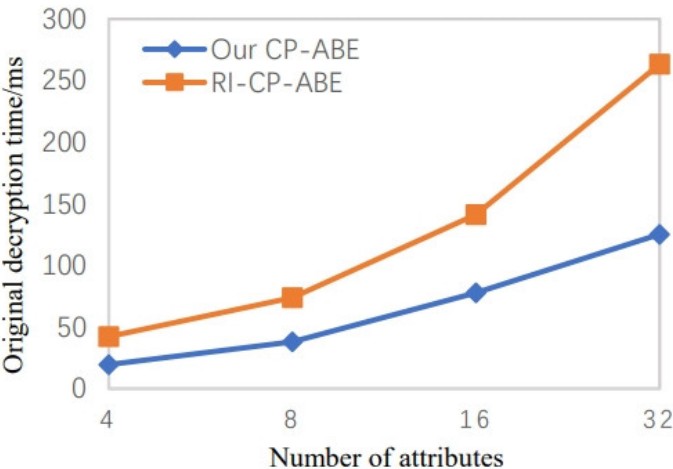

**Figure 6.** Original decryption time when the number of attributes increases [21].

The system revocation time and the decryption time after the user is revoked overhead are illustrated in Figures 7 and 8, respectively. Our scheme requires less time calculation in

the user revocation stage and the decryption step than RI-CP-ABE. As a result, our scheme has higher efficiency in practical applications.

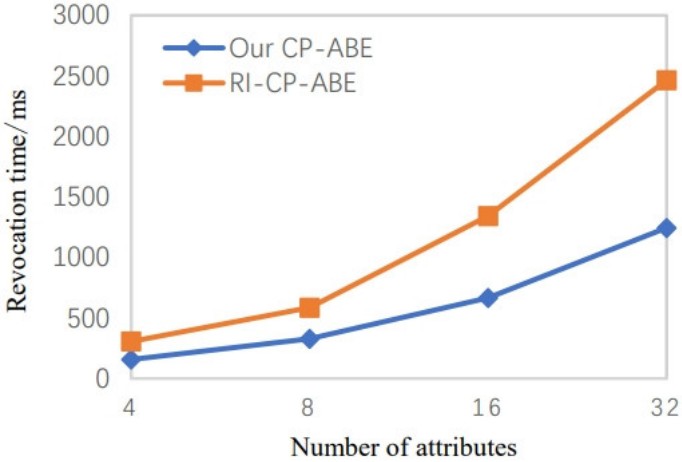

**Figure 7.** Revocation time when the number of attributes increases [21].

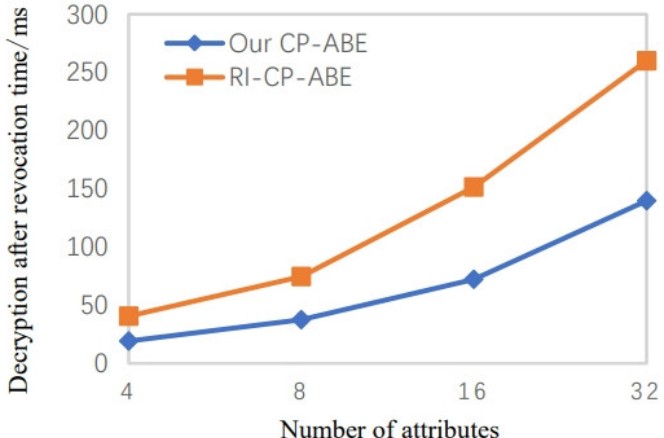

**Figure 8.** Decryption time after revoking user when the number of attributes increases [21].

## 7. Conclusions and Prospect

In this article, we construct an efficient RABE scheme that supports data integrity and solves the key escrow problem. User revocation is achieved using the ciphertext delegation algorithm, and the user can check whether the plaintext corresponding to the new ciphertext is the same as the original plaintext. Compared with the previous scheme with integrity verification, our scheme is more efficient. In addition, we introduced an attribute authority, and the key authority and attribute authority jointly generate private keys for users, which solve the key escrow issue effectively. Finally, the safety of the scheme is proved under the standard model and we give a performance analysis of our scheme. The scheme in this paper only supports the integrity verification under user revocation. Our next research will address the question of how to support the integrity verification under attribute revocation.

**Author Contributions:** Conceptualization and methodology, M.H.; software and validation, Y.Z. and M.Z.; writing—original draft preparation, Y.L.; writing—review and editing, M.H. and Y.L.; funding acquisition, M.H. and Y.Z.; proof reading, B.Y.; validation of results, Y.L. and M.Z. All authors have read and agreed to the published version of the manuscript.

**Funding:** This research was funded by the National Natural Science Foundation of China, grant number 62202375, the Natural Science Basic Research Program of Shaanxi Province, grant number 2021MJ-514 and 2022JQ-604, the Young Talent Fund of Association for Science and Technology in Shaanxi, China, grant number 20220134.

**Institutional Review Board Statement:** Not applicable.

**Informed Consent Statement:** Not applicable.

**Data Availability Statement:** No new data were created or analyzed in this study. Data sharing is not applicable to this article.

**Conflicts of Interest:** The authors declare that they have no known competing financial interests or personal relationships that could have appeared to influence the work reported in this paper.

**Appendix A**

To facilitate the readers' understanding, we give the algorithm flow of reference [21] as follows:

(1) $Setup(\lambda, U)$: The authority center generates a bilinear pairing tuple $(e, G, G_T, g, p)$. Chooses random value $g, h_1, h_2, \cdots, h_U, \phi, \varphi \in G, \alpha, a \in Z_p^*$ and a hash function $H : G_T \to Z_p^*$. Sets the master secret key $msk = g^\alpha$ and public parameters

$$PP = \left(e, G, G_T, g, h_1, \cdots, h_U, \phi, \varphi, g^a, e(g,g)^\alpha, H\right)$$

(2) $KeyGen(msk, Att)$: The authority center chooses a random value $s \in Z_p^*$, and computes $sk = \{Att, K = g^\alpha g^{as}, K_0 = g^s, \forall x \in Att, K_x = h_x{}^s\}$.

(3) $Enc(m, (M, f))$: On input a message m and an access policy $(M, f)$, $M$ is an $t \times k$ matrix and $f$ associates each row of $M$ to an attribute. The algorithm selects two random vectors $\overrightarrow{\mu} = (r, y_2, \cdots, y_k) \in Z_p^k$ and $\overrightarrow{v} = (\overrightarrow{r}, \overrightarrow{y}_2, \cdots, \overrightarrow{y}_k) \in Z_p^k$. For each row $M_j$ of $M$, computes $\lambda_j = \overrightarrow{\mu} \cdot M_j$ and $\overrightarrow{\lambda}_j = \overrightarrow{v} \cdot M_j, j \in [1, t]$. Randomly chooses $r_j, \overrightarrow{r}_j \in Z_p$ for each $j \in [1, t]$ and $m' \in G_T$. Then computes $C_1 = m \cdot e(g, g)^{\alpha r}$, $C_2 = g^r, C_{3,j} = g^{a\lambda_j} h_{f(j)}^{-r_j}, C_{4,j} = g^{r_j}, \forall j \in [1, t]$. $D_1 = m' \cdot e(g, g)^{\alpha \overline{r}}, D_2 = g^{\overline{r}}$, $D_{3,j} = g^{a\overline{\lambda}_j} h_{f(j)}^{-\overline{r}_j}, C_{4,j} = g^{\overline{r}_j}, \forall j \in [1, t], \overline{C} = \phi^{H(m)} \varphi^{H(m')}$. Outputs the ciphertext as $CT = ((M, f), C_1, C_2, C_{3,j}, C_{4,j}, D_1, D_2, D_{3,j}, D_{4,j}, \overline{C},), j \in [1, t]$.

(4) $Dec(sk, CT)$: On input a secret key $sk = \{Att, K, K_0, K_x\}$ and a ciphertext $CT = ((M, f), C_1, C_2, C_{3,j}, C_{4,j}, D_1, D_2, D_{3,j}, D_{4,j}, \overline{C},)$, the recipient first checks whether $R(Att, (M, f)) = 1$. If $R(Att, (M, f)) \neq 1$, outputs an error symbol $\perp$. Otherwise, finds the set $T \subset \{1, \cdots, t\}$ and $T = \{j : f(j) \in Att\}$. Computes constant element $\theta_j \in Z_p^*$, such that $\Sigma_{j \in T} \theta_j M_j = (1, 0, 0, \cdots, 0)$. Then the recipient computes

$$m = C_1 / \frac{e(K, C_2)}{(\Pi_{j \in T} e(K_0, C_{3,j}) \cdot e(K_{f(j)}, C_{4,j}))^{\theta_j}} \text{ and}$$
$$m' = D_1 / \frac{e(K, D_2)}{(\Pi_{j \in T} e(K_0, D_{3,j}) \cdot e(K_{f(j)}, D_{4,j}))^{\theta_j}}.$$

Checks if $\overline{C} = \phi^{H(m)} \varphi^{H(m')}$, outputs $m$. Otherwise outputs an error symbol $\perp$.

(5) $Revoke(CT, (\widetilde{M}, \widetilde{f}))$: On input a ciphertext $CT$ and a revocation access policy $(\widetilde{M}, \widetilde{f})$, where $M$ and $\widetilde{M}$ are $t \times k$ and $\widetilde{t} \times \widetilde{k}$ matrixes, outputs a revoked ciphertext for access policy $(M', f')$. Sets $(M', f')$ as

$$M' = \begin{pmatrix} M & -c_1|0 \\ 0 & \widetilde{M} \end{pmatrix}, f'(j) = \begin{cases} f(j), j \leq t \\ \widetilde{f}(j - t), j > t \end{cases},$$

where $c_1$ is the first column of $M$. Note that $M'$ is an $t' \times k'$ matrix, where $t' = t + \widetilde{t}$, $k' = k + \widetilde{k}$. Computes $C_1'' = C_1, C_2'' = C_2$,

$$\begin{cases} C_{3,j}'' = C_{3,j}, C_{4,j}'' = C_{4,j}, j \in [1, t] \\ C_{3,j}'' = 1_G, C_{4,j}'' = 1_G, j \in [t+1, t'] \end{cases}$$

where $1_G$ is the identity element of group $G$. Then selects a random vector $\vec{\mu}''' = (r''', y_2''', \cdots, y_k''') \in Z_p^{k'}$. For each row $M_j'$ of $M'$, computes $\lambda_j''' = \vec{\mu}''' \cdot M_j'$, $j \in [1, t']$. Randomly chooses $r_j''' \in Z_p$ for each $j \in [1, t']$. Then computes a random ciphertext $CT'''$ as

$$C_1''' = e(g,g)^{\alpha r'''}, C_2''' = g^{r'''}, C_{3,j}''' = g^{a\lambda_j'''} h_{f(j)}^{-r_j'''}, C_{4,j}''' = g^{r_j'''}, \forall j \in [1, t'].$$

Then, computes

$$C_1' = C_1'' \cdot C_1''', C_2' = C_2'' \cdot C_2''', C_{3,j}' = C_{3,j}'' \cdot C_{3,j}''', C_{4,j}' = C_{4,j}'' \cdot C_{4,j}''', \forall j \in [1, t'].$$

The value $D_1', D_2', D_{3,j}', D_{4,j}', j \in [1, t']$ can be computed in the same manner. Sets $\overline{C}' = \overline{C}$. Finally, outputs the revoked ciphertext

$$CT' = \left( (M', \rho'), C_1', C_2', C_{3,j}', C_{4,j}', D_1', D_2', D_{3,j}', D_{4,j}', \overline{C}', j \in [1, t'] \right).$$

(6) $Dec_{re}(sk', CT, CT')$: On input a secret $sk'$ of attribute set $Att'$, an original ciphertext $CT = \left( (M, f), C_1, C_2, C_{3,j}, C_{4,j}, D_1, D_2, D_{3,j}, D_{4,j}, \overline{C}, \right)$ and a revoked ciphertext $CT' = \left( (M', \rho'), C_1', C_2', C_{3,j}', C_{4,j}', D_1', D_2', D_{3,j}', D_{4,j}', \overline{C}' \right)$, it verifies whether $\overline{C}' = \overline{C}$. If not, outputs an error symbol $\perp$ and abort. Then, it checks whether $R(Att', (M', f')) = 1$. If $R(Att', (M', f')) \neq 1$, outputs an error symbol $\perp$ and abort. Otherwise, finds the set $T' \subset \{1, \cdots, t'\}$ and $T' = \{j : f'(j) \in Att'\}$. Computes constant element $\theta_j' \in Z_p^*$, such that $\Sigma_{j \in T'} \theta_j' M_j' = (1, 0, 0, \cdots, 0)$. Then, it computes

$$m = C_1' / \frac{e(K, C_2')}{(\Pi_{j \in T'} e(K_0, C_{3,j}') \cdot e(K_{f'(j)}, C_{4,j}'))^{\theta_j'}},$$

$$m' = D_1' / \frac{e(K, D_2')}{(\Pi_{j \in T'} e(K_0, D_{3,j}') \cdot e(K_{f'(j)}, D_{4,j}'))^{\theta_j'}}.$$

Checks if $\overline{C}' = \phi^{H(m)} \varphi^{H(m')}$, outputs $m$. Otherwise outputs an error symbol $\perp$.

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
