# Peer review of "Efficient Revocable Attribute-Based Encryption with Data Integrity and Key Escrow-Free"

_information, doi:10.3390/info15010032_

Round 1

Reviewer 1 Report (Previous Reviewer 1)

Comments and Suggestions for Authors

I recommend paper's acceptance now after observing that all my concerns are addressed

Comments on the Quality of English Language

The English level is acceptable.

Reviewer 2 Report (Previous Reviewer 2)

Comments and Suggestions for Authors

I have no additional comments and recommend publication.

This manuscript is a resubmission of an earlier submission. The following is a list of the peer review reports and author responses from that submission.

Round 1

Reviewer 1 Report

Comments and Suggestions for Authors

In this paper, authors present an efficient revocable ABE scheme that supports data integrity and solves the key escrow issue. Performance analysis illustrated the  practicability and effectiveness of the proposed scheme with respect to previous works [21]. In general, the paper is well written and the  organization structure is also in good condition. The contributions of the paper are also suitable for publications. I recommend paper's acceptance.

Some  comments:

- Simulation parameters used in experiments should be given?

- How fair are comparisons with [21] in simulations? How are parameters selected to make fair comparisons with different scheme?

- A notation table is required

- On page 10 line 378, what is the symbol next to PPT adversary? 

Comments on the Quality of English Language

The English level is acceptable

Reviewer 2 Report

Comments and Suggestions for Authors

The manuscript presents a scheme that may be a slight improvement over the similar schemes it references, specifically the one in Reference 21. The authors do provide the requisite proofs for their theorems. However, the main advantage the authors cite is the Key-escrow free property in table 2, which in terms of importance falls below Integrity and User Revocation. It is not clear how the results in figures 3 through 8 were computed. Paragraph 6.2 defines two attributes, s, and u. The x-axis in figures 3 through 8 is simply 'number of attributes'. And it is not clear how the authors apply the Table 3 relationships to the approach in [21]. A short appendix would be helpful to convince the reader that (6u+4)E+2P applied to the scheme in [21] is a fair comparison to (3u+3)E+P  for the author's scheme.

Reviewer 3 Report

Comments and Suggestions for Authors

The paper discusses the limitations of current revocable attribute-based encryption (RABE) schemes, particularly in terms of data integrity and the key escrow issue. It proposes an efficient RABE scheme that addresses these issues and provides security based on the decisional q-parallel bilinear Diffie-Hellman exponent (q-PBDHE) assumption and discrete logarithm (DL) assumption. The performance analysis demonstrates the efficiency of the proposed scheme.

The research problem is clearly defined, and the paper's contributions are well-described and demonstrated. The steps of the proposed RABE system are shown in detail and in clear visual form.  Intense mathematical expressions have been utilised to describe the algorithms in the RABE scheme, the security model, the proposed key issuing protocol, and proofing theorems 1 and 2 in security analysis. The performance of the proposed scheme is analysed clearly and critically in terms of functionality, computational cost, and experimental perspectives.

The paper's results should be expounded upon in the conclusion to make it more comprehensive. Future works should also be included.
